# Determination of the Ecotoxicity of Herbicides Roundup^®^ Classic Pro and Garlon New in Aquatic and Terrestrial Environments

**DOI:** 10.3390/plants9091203

**Published:** 2020-09-14

**Authors:** Lucia Tajnaiová, Radek Vurm, Marina Kholomyeva, Miroslav Kobera, Vladimír Kočí

**Affiliations:** Faculty of Environmental Technology, Department of Environmental Chemistry, UCT Prague, Technická 5, 166 28 Prague, Czech Republic; vurmr@vscht.cz (R.V.); kholomym@vscht.cz (M.K.); miroslavkobera@gmail.com (M.K.); kociv@vscht.cz (V.K.)

**Keywords:** glyphosate, triclopyr, AMPA, *Desmodesmus subspicatus*, *Lemna minor*, ecotoxicity, herbicides, Roundup, Garlon, dehydrogenase activity

## Abstract

Herbicides help increase agricultural yields significantly, but they may negatively impact the life of non-target organisms. Modifying the life cycle of primary producers can affect other organisms in the food chain, and consequently in the whole ecosystem. We investigated the effect of common herbicides Roundup^®^ Classic Pro (active substance glyphosate) and Garlon New (triclopyr and fluroxypyr) on aquatic organisms duckweed *Lemna minor* and green algae *Desmodesmus subspicatus*, and on the enzymatic activity of soil. We also compared the effects of Roundup^®^ Classic Pro to that of a metabolite of its active substance, aminomethylphosphonic acid (AMPA). The results of an algal growth test showed that AMPA has a 1.5× weaker inhibitory effect on the growth of *D. subspicatus* than the Roundup formula, and the strongest growth inhibition was caused by Garlon New (IC_50_Roundup = 267.3 µg/L, IC_50_Garlon = 21.0 µg/L, IC_50_AMPA = 117.8 mg/L). The results of the duckweed growth inhibition test revealed that Roundup and Garlon New caused 100% growth inhibition of *L. minor* even at significantly lower concentrations than the ready-to-use concentration. The total chlorophyll content in the fronds was lowest when Garlon New was used. The highest dehydrogenase activity was observed in soil treated with Garlon New, and the lowest in soil treated with Roundup^®^ Classic Pro. The results of this study showed that all three tested substances were ecotoxic to the tested organisms.

## 1. Introduction

Pesticides, often referred to as plant protection products, are chemical substances that are widely used in agriculture and forestry [1,2]. Among the most important groups of these substances are herbicides, which are used to control the growth of weeds.

The most widely used herbicides are the organophosphate-based herbicides, Roundup [3], whose active ingredient is isopropylamine salt of glyphosate [4], and Garlon [5], the active ingredient being triclopyric acid, belonging to the class of pyridinecarboxylic acids. Glyphosate (N-(phosphonomethyl)glycine) is a non-selective weed-killer [6,7] which effectively inhibits the 5-enolpyruvyl-shikimate-3-phosphate synthase (EPSPS) enzyme of the shikimate pathway [8]. This enzyme is involved in the production of aromatic amino acids (phenylalanine, tyrosine, and tryptophan), which are essential for plant growth [9]. This enzyme is also found in other organisms, such as bacteria and algae, which are not the primary targets of herbicides [10], but it is not found in mammals [11]. Nevertheless, some studies suggest that glyphosate-based herbicides are toxic to non-target organisms. According to Gasnier et al., such herbicides are toxic to human cells and act as endocrine disruptors, even at doses that are much lower than those used in agriculture [12]. Regarding genotoxicity, studies differ on the effects of Roundup. According to Cavalcante et al., Roundup causes genotoxic damage to the erythrocytes and gill cells of the fish *Prochilodus lineatus* [3]. It has been demonstrated that glyphosate has a negative impact on trees (*Handroanthus chrysotrichus*, *Garcinia gardneriana*) [13], spiders (*Neoscona theisi*) [14], (*Pardosa agrestis*) [15], lizards (*Podarcis siculus*) [16], as well as earthworms (*Eisenia fetida*), for which the results suggest that exposure is sublethal [17]. For two types of earthworms (*Drawida willsi* and *Lampito mauritii*), high concentrations of glyphosate have been found to cause lesions, skin undulation, and altered biochemical parameters [18]. On the other hand, no negative effects were observed in a population of springtails (*Isotomidae* sp.) exposed to glyphosate, and the population number even increased with increasing herbicide concentration (0, 2, and 4 L/ha) [19].

In soil, glyphosate has a relatively short half-life, ranging from eleven to seventeen days [20]. Glyphosate can be degraded in soil by two main pathways. The first is its conversion to sarcosine and inorganic phosphate via C-P lyase. The second and more common route is the degradation of glyphosate to aminomethylphosphonic acid (AMPA) and glyoxylate, which is caused by the glyphosate oxidoreductase (GOX) enzyme of the class that cleaves the C-N bond of glyphosate [21,22,23]. AMPA is a weak phosphate-based acid, which is soluble in water [24]. AMPA is widespread in groundwater [25], sediments [26], and arable land where it decomposes slowly [27]. Comet assay tests on zebrafish (*Danio rerio*) larvae cells showed that both active substances glyphosate and its metabolite AMPA, were equally genotoxic to them [28]. Survival and reproduction tests on earthworms (*E. fetida*), mite (*Hypoaspis aculeifer*), and springtails (*Folsomia candida*) exposed to glyphosate and AMPA showed no significant differences compared to the control group [29]. Several studies suggest that there is no significant difference between the effects of glyphosate and AMPA on different organisms, but that both substances are less toxic than Roundup itself [11,30,31]. The reason for this could be the fact that Roundup includes in its formula a polyoxyethylene amine (POEA) adjuvant, a surfactant that increases the effectiveness of the herbicide [30]. According to Bonansea et al., whose work was aimed at estimating the environmental risk associated with the presence of glyphosate and AMPA in crop areas, there is a risk to aquatic organisms associated with the presence of glyphosate in water [32].

Triclopyr (3,5,6-trichloro-2-pyridinyloxyacetic acid) is a selective and systemic herbicide with an auxin-type mechanism of action. Plants receive triclopyr through their roots, stems, and leaf tissues. Species susceptible to this chemical die within seven to fourteen days of its application [33]. Results from field studies show that triethlyamine salt of triclopyr and its metabolites, 3,5,6-trichloro-2-pyridinol (TCP) and 3,5,6-trichloro-2-methoxy-pyridine (TMP), dissipate in water at a somewhat similar rate (ranges of 0.5 to 7.5 days, 4.2 to 10.0 days, and 4.0 to 8.8 days, respectively) [34]. Triclopyr has been shown to be toxic in small doses to non-target organisms, such as plants yarrow (*Achillea millefolium* L.) and fireweed (*Chamerion angustifolium* L.) [35], insects *Epeorus vitreus*, *Dolophilodes distinctus*, *Isogenoides* sp., and *Hydropsyche* sp. [36], and genotoxic to a critically endangered eel (*Anguilla anguilla* L.) [37]. The exposure of red-legged frogs (*Rana aurora*) to triclopyr did not lead to increased mortality, organ damage, or changes in post-exposure behavior; although, lethargy during exposure and a prolongation of the metamorphosis cycle by one day were observed [38]. No effects were observed on the survival of tadpoles of the northern leopard frog (*Lithobates pipiens*); although, they were physically smaller after the exposure to triclopyr [39]. Tests on embryos and tadpoles of the green frog (*Rana clamitans*), leopard frog (*Rana pipiens*), and bull frog (*Rana catesbeiana*) show that triclopyr has no effect on the hatching, behavior, or size of these amphibians, yet freshly-hatched individuals are very sensitive to the herbicide, and increased mortality or paralysis may occur [40].

Fluroxypyr (([4-Amino-3,5-dichloro-6-fluoro-2-pyridinyl]oxy)acetic acid) is a pyridine herbicide which has the potential to contaminate groundwater resources [41]. This auxin-type herbicide causes an imbalance in auxin homeostasis and triggers a subsequent series of physiological and biochemical processes [42]. So far, the photodegradation of triclopyr in drinking and leachate water has been shown to be significantly lower than that of fluroxypyr [43].

*D. subspicatus* is an eukaryotic organism that responds to inappropriate growth conditions or to the presence of toxic contaminants at the cellular level by rapidly increasing its production of heat shock proteins [44,45]. Green algae generally, and *D. subspicatus* in particular, play an important role in the aquatic food chain [46]; therefore, any herbicide intoxication would not only affect them, but the whole ecosystem.

*L. minor* is a floating vascular plant [47] that is often used as a bioindicator of pollutants, such as heavy metals [48,49] or pesticides [50]. Many studies focus on the possibility of using duckweed as a plant suitable for phytoremediation of metals [51,52,53] or organic pollutants, such as 1,4-dioxane [54].

Herbicides may affect the biological properties of the soil [55]. Soil enzyme activity is an indicator of processes in the soil environment, including metabolic processes, and levels of contamination [56]. According to Sebiomo et al., the treatment of the soil with active substances of herbicides (atrazine, paraquat, and glyphosate) results in a decrease of dehydrogenase activity (DHA) when compared to the control group [57].

The primary aim of this work was to compare the effects of two commonly available herbicides, Roundup^®^ Classic Pro and Garlon New, the former with glyphosate content and the latter without it, on non-target aquatic organisms, namely common duckweed (*L. minor*) and green algae *D. subspicatus*, which are an important part of the food chain. Garlon New was chosen for its excellent availability on the Czech market, where it is one of the best-selling plant protection products. Its effect should be apparent several hours after application, which is suitable for acute toxicity tests. Part of this analysis was measuring the effect of these herbicides on enzymatic activity in soil. An additional aim was to compare the effects of the glyphosate-containing herbicide Roundup^®^ Classic Pro to the effects of AMPA, a metabolite of glyphosate. Changes in the growth rates of two tested organisms, the total chlorophyll content of duckweed fronds, and the levels of DHA in soil were all analyzed.

## 2. Materials and Methods

### 2.1. Chemicals

Three different chemical substances were tested: Roundup^®^ Classic Pro, Garlon New, and aminomethylphosphonic acid (AMPA). The commercial glyphosate-based herbicide in this study was Roundup^®^ Classic Pro manufactured by Monsanto Canada, Ottawa. The main substance content was 28.85% w/v of glyphosate. The additive surfactant (ether alkylamine ethoxylate) comprised 6%, and the rest was a combination of water and small amounts of other chemicals. The second commercially available herbicide used in this work was Garlon New, oil-soluble, water-emulsifiable butoxyethyl ester of triclopyr (3,5,6-trichloro-2-pyridinyloxyacetic acid) supplied by AgroBio Opava, s.r.o. Garlon New consists of 8.23% *w*/*v* triclopyr-triethylammonium and 2.83% fluroxypyre MHE, with the rest of the formulation being inert carbohydrates and alcohols. AMPA was supplied by Tokio Chemical Industry co. Ltd., and was used to prepare a stock solution.

For aquatic tests, Bold’s basal medium (BBM), pH 6.5 [58] and Steinberg growth medium, pH 5.5 were prepared. Methanol was used to determine chlorophyll content. For the determination of DHA in soils, analytical grade ethanol (96%), hydrochloric acid (HCl, c = 1 mol·L^−1^), tris buffer solution (c = 100 nmol·L^−1^, pH = 7.6), substrate solution (TTC, c = 300 nmol·L^−1^), and triphenylformazan solutions (TPF) were used. All solutions for the determination of DHA were prepared according to ISO 23753-1.

### 2.2. Organisms

Two aquatic organisms were chosen for this experiment. The first one was green alga *Desmodesmus subspicatus* (R. Chodat), E. Hegewald *et* A. Schmidt, Brinkmann 1953/SAG 8681 BU AV Czech Republic, Třeboň. The second one was duckweed *Lemna minor* L., sterile culture, Steinberg: Federal Environmental Agency (FDA), Berlin, Germany.

### 2.3. Experimental Design

The experimental design was carried out according to the following standards: EN ISO 8692 [59], EN ISO 20079 [60], EN ISO 23753-1 [61], with modifications.

#### 2.3.1. Algal Growth Inhibition Test

The algal growth inhibition test was performed in 25 mL Erlenmeyer flasks. Pre-cultivation of green algae *D. subspicatus* in a 500 mL flat bottom flask was started three days prior to the inhibition test. Throughout the pre-cultivation, the flask was aerated by a tube with a microbiological filter. The cells were inoculated into the growth medium and subsequently cultured at 23 ± 2 °C under continuous illumination provided by daylight lamps with an intensity of 8000 lx under sterile conditions. Test substances in different concentrations, BBM, and inoculum of a total volume of 15 mL were pipetted into the Erlenmeyer flasks under sterile conditions in a laminar flow box (BIO190 CYTO A2, Alpina, Konin, Poland). Subsequently, the flasks were stoppered with pulp plugs and cultured (Sanyo MLR-351, Sanyo Electric Co., Moriguchi, Japan) under the same heat and light conditions as during pre-cultivation. The initial algal density of *D. subspicatus* depended on the concentration of the algal inoculum. During the test, the flasks were continuously shaken at 160 rpm by an orbital shaker (ELMI DOS-20L, ELMI, Calabasas, California, USA). After 72 h, the cell concentration of *D. subspicatus* was determined with a light microscope (Leica DM750, Leica Microsystems, Wetzlar, Germany) at 400× magnification, using Bürker counting chambers. The cells were counted in 24 squares—on the two diagonals. All vessels and equipment used in the test were made of glass or other chemically inert material. To compensate for pH changes in the algae test media after the addition of the AMPA solution, the pH was adjusted. Before the application of testing organisms and the sterilization, the pH was changed using NaOH (c = 0.1 mol·L^−1^) to pH = 6.6, which is the standardized pH of an algae cultivation medium. Roundup^®^ Classic Pro, Garlon New, and AMPA were tested in 15 concentrations, complemented by control samples. The highest concentration of the herbicides tested was their ready-to-use concentration. Specifically, the following concentrations were used for Roundup^®^ Classic Pro: (18,381; 13,785.7; 9190.5; 4595.2; 2297.6; 1148.8; 574.4; 287.2; 143.6; 71.8; 35.9; 17.9; 8.9; 4.4; and 2.2) µg.·L^−1^. For Garlon New, the following concentrations were used: (15,000; 11,250; 7500; 3750; 1875; 937.5; 468.7; 234.3; 117.1; 58.5; 29.2; 14.6; 7.3; 3.6; and 1.8) µg·L^−1^. For AMPA, the concentrations which were used corresponded to the molar equivalent of glyphosate present in the respective dilutions of Roundup^®^ Classic Pro. The highest concentration used was 2851 mg·L^−1^ and other values used were as follows: (2138.6; 1425.7; 712.8; 356.4; 178.2; 128.3; 85.5; 64.1; 42.7; 21.3; 10.6; 5.3; 2.6; and 1.3) mg·L^−1^. In the pH-adjusted assay, decimal dilution was used and five concentrations plus a control were tested (2851.5; 285.1; 28.5; 2.8; and 0.2) mg·L^−1^. To determine the acute toxicity of each of the tested substances after 72 h, their IC_50_ values were calculated using the endpoint of growth inhibition. The specific growth rate was calculated as the logarithmic increase in biomass according to the following equation:(1)$$µ=\frac{\ln c\left(e\right)-\ln c\left(s\right)}{\Delta t}$$
where µ is the average growth rate in h^−1^, c(e) is the biomass concentration at the end of the test, c(s) is the biomass concentration at the beginning of the test and Δt is the difference in time between the beginning and the end of the test in h. In our case, the initial value of biomass was 80,000 cells·mL^−1^.The inhibition of growth rate in percent was calculated using the results of the previous equation as follows:(2)$$I=\frac{µ\left(c\right)-\;µ\left(s\right)}{µ\left(c\right)}\;\times 100$$
where I is the inhibition of specific growth rate in percentage, µ(c) is the growth rate for control sample in h^−1^, µ(s) is the growth rate for the test sample in h^−1^.

#### 2.3.2. Duckweed Growth Inhibition Test

The duckweed growth inhibition test was initiated by pre-cultivating *L. minor* in the Steinberg medium, whose pH was adjusted using NaOH (c = 0.1 mol·L^−1^) and HCl (c = 1 mol·L^−1^) to pH = 5.5 ± 0.2, for 7 days. Prior to transferring *L. minor* to the pre-culture chamber, the culture was kept in a sterile environment, so as not to be contaminated with bacteria, and was held under reduced illumination and a temperature of 6 °C. After the adaptation period, several healthy colonies with 2–3 fronds (12 fronds per beaker total) were transferred from the inoculum culture into 150 mL glass beakers containing 100 mL of modified Steinberg medium used for dilution of the prepared ready-to-use herbicide solution. All beakers were covered and placed in a cultivator (WTW tr-1, WTW, Prague, Czechia), where the cultures were grown under continuous cool fluorescent light at 24 ± 1 °C. All fronds of *L. minor* were equidistant from the light (7000 lux). The conditions throughout the entire test were sterile. The test with the AMPA solution was complemented with a parallel test using a pH-adjusted AMPA solution of pH = 5.5, the same pH as the modified Steinberg medium.

Duckweed growth was determined by tallying the frond numbers (FN) and the area of the fronds. The frond numbers were taken at the first, fourth, and seventh day of the experiment. All visible fronds were counted. A photo of the surface of the fronds was taken simultaneously for area evaluation using the NIS-Elements 4.2 software (Laboratory Imaging). Fronds of *L. minor* were observed at the first, fourth, and seventh day of the experiment for toxicity symptoms, such as chlorosis, necrosis, and frond disconnection.

Roundup^®^ Classic Pro, Garlon New, and AMPA were tested in 10 concentrations plus a control. The highest concentration of test substances was their ready-to-use concentration. The full list of concentrations for Roundup^®^ Classic Pro is as follows: (18,381; 13,758.7; 9190.5; 4595.2; 2297.6; 1148.8; 574.4; 287.2; 143.6; and 71.8) µg·L^−1^. The following concentrations were tested for Garlon New: (15,000; 11,250; 7500; 3750; 1875; 937.5; 468.7; 234.3; 117.1; and 58.5) µg·L^−1^. For AMPA, 10 concentrations plus a control were used, each of which corresponded to the molar equivalent of glyphosate present in its respective concentration of Roundup^®^ Classic Pro. The highest concentration used was 2852 mg·L^−1^ and the dilutions were the same as for the tested herbicides. This means that the AMPA concentrations tested were as follows: (2851.6; 2138.7; 1425.8; 712.9; 356.4; 178.2; 85.5; 42.7; 21.3; and 10.6) mg·L^−1^. In the pH-adjusted assay, decimal dilution was used and 5 concentrations plus a control were tested: (2851.6; 285.1; 28.5; 2.8; and 0.2) mg·L^−1^.

The specific growth rate for was calculated as the logarithmic increase according to the following equation:(3)$$µ=\frac{\ln c\left(e\right)-\ln c\left(s\right)}{\Delta t}$$
where µ is the average growth rate in d^−1^, c(e) is the frond area at the end of the test in mm^2^, c(s) is the frond area at the beginning of the test in mm^2^, and Δt is the difference in time between the beginning and the end of the test in d. The inhibition of growth rate in percent was calculated using the results of the previous equation as follows:(4)$$I=\frac{µ\left(c\right)-µ\left(s\right)}{µ\left(c\right)}\;\times 100$$
where I is the inhibition of specific growth rate in percentage, µ(c) is the growth rate for the control sample in d^−1^, and µ(s) is the growth rate for the test sample in d^−1^. To verify the validity of the test, the doubling time of the frond area was calculated according to the equation:(5)$$T=\frac{\ln2}{µ}$$
where µ is the average growth rate.

Chlorophyll *a* (chl *a*), *b* (chl *b*) concentrations were determined by extraction in methanol. After 7 days of incubation, duckweed fronds were gathered and placed in a plastic vial. The roots of the fronds were removed. Next, the plastic vials were filled up with 8 mL of methanol. The extraction was carried out in a cold environment (4 °C) and darkness [62]. After 24 h of extraction, the samples were centrifuged at 5000 rpm for 15 min and then a spectrophotometric analysis of the supernatant was performed.

To measure the concentration of chlorophyll *a* and *b*, the absorbance of the samples was measured at different wave lengths. Following the spectra of purified chlorophyll, *a* and *b*, peak maxima were determined to be 666 nm and 653 nm, respectively (Spectrophotometer Hach DR/2400, Hach Company, Loveland, CO, USA).

#### 2.3.3. The Determination of Dehydrogenase Activity in Soil

A sample of soil was taken according to ČSN ISO 10381-6 [63] from a depth of 0–15 cm in Dejvice, Prague (Czech Republic, coordinates: 50°06′09.3″ N 14°23′16.2″ E). It is a loess aluminosilicate soil with an admixture of silicates on slate subsoil. After having been sieved through a 2 mm sieve, the soil was placed in a resealable plastic bag and stored in a refrigerator in the dark at 4 °C for 7 days. Prior to the DHA test, the pH was determined according to ČSN ISO 10390 [64]. The soil suspension was prepared by mixing the sample with five times its volume of distilled water and then shaking it without access to oxygen for one hour. The moisture content of the soil was gravimetrically measured according to ČSN ISO 11465 [65] by drying 100 g of a soil sample at 105 °C (dryer UF 110, Memmert, Buechenbach, Germany), and then reweighed on analytical balances (ALJ 500–4A, Kern, Balingen, Germany). After an initial oven drying at 105 °C for water loss, the organic matter content of the soil was determined by the loss on ignition (LOI) method. The sample was ignited in a muffle laboratory furnace (LM 212, VEB Electro, Bad Frankenhausen, Germany) for 4 h at 550 °C. The furnace had been refurbished and specially adapted for thermodesorption experiments, equipped with Clare 4.0 control (Classic). The samples were weighed before and after the analysis. The content of elements C, H, N, and S in the soil was quantified by burning the sample at 1200 °C (Elementar Vario EL Cube with TCD detection, Elementar, Langenselbold, Germany). The gaseous combustion products (N_2_, CO_2_, H_2_O, and SO_2_) were purified, separated into individual components, and analyzed on a TCD detector. The results of the analysis include all of the combustible sulfur, both organic and inorganic, as well as all of the combustible carbon, both organically bound and inorganically bound. The content of the elements was determined in two replication measurements. 4-amino-benzene-sulfonic acid (Sigma Aldrich) was used as a standard.

The first combination of the concentrations to be used was calculated as per label instructions, where the usage is stated at 30 mL in 2 L of water per 100 m^2^ of agricultural soil. The label guidance was identical for both chosen herbicides, Roundup^®^ Classic Pro and Garlon New alike. After a conversion to the area and soil volume in the test tubes, a volume of 50 μL of each test substance was used. In our study, the effect of AMPA on DHA was compared to the effect of the herbicide Roundup on DHA. For the comparison to be informative, the dosage of AMPA was derived from the glyphosate content in Roundup as opposed to Roundup as a whole. The concentration of glyphosate salt in the Roundup herbicide was stated to be 360 g·L^−1^, which in turn means that its ready-to-use solution contains 5.4 g·L^−1^ of glyphosate when prepared according to label instructions, i.e., 15 mL·L^−1^ of Roundup. The highest AMPA concentration corresponded to the molar equivalent of glyphosate salt present in the Roundup formula, which is 2852 mg·L^−1^. Rationed for our purposes, 50 μL of each herbicidal solution was used per two grams of soil. This means that at 100% concentration, we ended up with 135 µg of glyphosate per gram of soil. Finally, we plotted the comparison between AMPA and Roundup onto a scale of dilutions corresponding to the highest concentration of Roundup and the equivalent highest concentration of AMPA. All glyphosate salt concentrations in the test medium at all dilutions are listed in a table in the Appendix A. The dehydrogenase activity when AMPA was applied, with all tested concentrations being molar equivalents of the respective glyphosate salt dilution levels, was higher at all dilutions than when Roundup was applied.

The soil was homogenized in a beaker and put in U-bottom tubes with 5 different concentrations of the herbicide, three per each concentration. The method used for measuring DHA in soil required that a 1% solution of 2,3,5-triphenyltetrazolium chloride (TTC) as an artificial electron acceptor be added to the soil samples. Tubes with stoppers filled with 2 g of soil were covered with aluminum foil and placed in an incubator (MBT 250, Kleinfeld, Gehrden, Germany) to be incubated in the dark at 25 °C for 24 h. Subsequently, methanol was added, and the tubes were vortexed (VELP Scientifica TX4, Usmate Italy) to better homogenize the sample. The samples were then centrifuged (Rotanta 460R, Hettich, Tuttlingen, Germany) for 10 min (2000 g, 20 °C) and the supernatant was pipetted off. Finally, the absorbance was measured by a UV-VIS spectrophotometer at 485 nm (Hach DR/2400, Hach Company, Loveland, CO, USA), and the concentration of triphenylformazane was established.

### 2.4. Statistical Analysis

All samples and controls were prepared in triplicate. The results are represented as the mean values ± their respective standard deviations. A one-way analysis of variance (ANOVA) with a post hoc Tukey–Kramer multiple comparison test (*p* = 0.05) were performed to determine whether the differences in the growth inhibition effects of the compared herbicides were significant. Alternatively, a one-way non-parametric ANOVA (Kruskal–Wallis) test with a post hoc Dunn’s multiple comparison test was performed for non-parametric data. All statistical analyses were performed with the Data Analysis Tools in Microsoft Excel 2016. The results of the statistical analyses are provided in the Appendix A. The standard deviations of the experimental data were calculated by means of the Microsoft Excel software. The half maximal inhibitory concentration (IC_50_) was calculated using GraphPad Prism 8.4.3.

## 3. Results and Discussion

### 3.1. Green Algae Acute Toxicity Test

The value of IC_50_ for the Roundup^®^ Classic Pro solution was found to be 267.3 µg·L^−1^. In the case of AMPA, a direct metabolite of glyphosate, the half maximal inhibitory concentration (IC_50_) was higher than that of the herbicide Roundup^®^ Classic Pro, namely IC_50_ = 117.8 mg·L^−1^. In contrast, for the herbicide Garlon New, the IC_50_ value was much lower, i.e., 21.0 µg·L^−1^. In summary, AMPA alone is less toxic to green algae than Roundup^®^ Classic Pro, but Garlon New inhibited the growth of *D. subspicatus* approximately 13-fold more than Roundup and much more than pure AMPA in our test of acute toxicity. Therefore, the leaching of even moderate amounts of Garlon New might have greater effects on aquatic freshwater algal species than the leaching of Roundup^®^ Classic Pro or its metabolite AMPA. After adding the test substances into the BBM medium, the pH of the medium decreased. AMPA caused the most significant change in the pH (Appendix A), so we proceeded to adjust the pH of a set of AMPA solutions. In the pH-adjusted solutions, the value of IC_50_ of AMPA was determined to be 192.1 mg·L^−1^, indicating a mild mitigation of its toxic effects. A pH-adjusted solution better represents in situ conditions where a pH buffering system, which would prevent the pH from decreasing in such an extreme way, is naturally present. The results of the toxicity tests are presented in Figure 1 and Figure 2. To make the charts more readable, there is a table in the Appendix A showing the inhibition values corresponding to all of the tested concentrations.

Figure 1 shows the inhibition caused by herbicides Roundup^®^ Classic Pro and Garlon New to green algae *D. subspicatus*. Roundup^®^ Classic Pro caused 85.1% inhibition at the ready-to-use concentration of 18,381 µg·L^−1^. Garlon New inhibited algae growth even more; the inhibition was calculated at 91.9% at the ready-to-use concentration of 15,000 µg·L^−1^. The observed algal cells that survived were smaller in size than those observed in the control. When the tested concentrations were reduced to 75% of the ready-to-use concentration, there was almost no difference in inhibition compared to the 100% dose for both herbicides, and their inhibitions were 84.3% and 90.4%, respectively. There was also no significant difference in the change in inhibition even with the 50% dilution of the ready-to-use dose. At the following doses tested, which were each half as concentrated as the previous one, the inhibition decreased only very slowly. It was still the case that the inhibition effect was stronger with Garlon New at all concentrations. A more significant decrease in the toxicity of Roundup^®^ Classic Pro occurred at the dose of 143.6 µg·L^−1^, which is 0.7% of the ready-to-use dose, but the inhibition was still higher than 50%, namely 63.8%. However, at this concentration, there was still no marked decrease in inhibition with Garlon New, as this remained at 90%. The content of the Roundup^®^ Classic Pro herbicide in the medium stopped inhibiting cell growth only as the tested dose reached as little as 4.4 µg·L^−1^, which corresponds to 0.02% of the recommended ready-to-use dose. At a dose of 2.2 µg·L^−1^, a growth stimulation of 5.8% was observed. The herbicide Garlon New did not stop inhibiting algal growth even at the lowest concentration tested, which corresponded to 0.01% of the ready-to-use dose. The real-world environment mitigates the effects of these herbicides. Nevertheless, if they cause such remarkable inhibition in laboratory tests at very low doses, we can assume that these substances can have a negative effect on non-target organisms even if natural conditions interfere.

Figure 2 shows the inhibition of *D. subspicatus* growth caused by AMPA and pH-adjusted AMPA, respectively. The inhibition caused by AMPA in the pH-adjusted medium at a dose corresponding to the ready-to-use dose was higher than that caused by AMPA without pH adjustment. At the highest dose tested (2851.6 mg·L^−1^), the inhibition caused by AMPA with and without pH adjustment was 92.8% and 89.6%, respectively. Inhibition values corresponding to AMPA without pH adjustment were in the range of 87.0%–89.6% for the first five most concentrated doses tested. The first major difference in the inhibition effect was observed at the dose of 128.3 mg·L^−1^ where inhibition dropped to 67.2%, and at the next dose of 85.5 mg·L^−1^, where inhibition was measured at 6.2%. Stimulation of growth occurred in AMPA without pH adjustments at a concentration of 21.3 mg·L^−1^. On the contrary, in AMPA with pH adjustment, inhibition effects persisted even at the concentration of 0.2 mg·L^−1^, which is over 10,000 times more diluted than the highest tested dose.

These results correspond with those of the Kruskal–Wallis post-hoc (Dunn’s) test, in which the pairwise comparison was performed to reveal the difference in the growth inhibition of *D. subspicatus*. Based on the obtained results, there was no difference between glyphosate and AMPA (*p* < 0.017); however, the difference between the pairs glyphosate-triclopyr and AMPA-triclopyr was significant (*p* > 0.017), (Appendix A).

Our results are in accordance with the criteria for the validity of the test according to the OECD guidelines for freshwater algae. The coefficient of variation of the daily growth rates in the control cultures during the test did not exceed 35% and the coefficient of variation of the average growth in the replicated control cultures did not exceed 15%. The pH of the control samples increased from the original 6.6 value in all three replicates, but it never exceeded the value of 8.0.

For AMPA, according to European Food Safety Authority (EFSA), the acute E_b_C_50_ endpoint for a biomass of green algae *D. subspicatus* is 89.8 mg·L^−1^ and the E_r_C_50_ for the growth rate is 452 mg·L^−1^ [66].

Our conclusions regarding glyphosate correspond to those by Nagai et al., who established that for glyphosate acting on the environment of *D. subspicatus*, EC_50_ is greater than 32 mg·L^−1^ [67]. The effects of commercial Roundup^®^ Classic Pro and pure glyphosate were also evaluated in terms of population growth, which, as in our work, was determined by counting the number of cells of green algae *Scenedesmus acutus*, *Scenedesmus quadricauda*, *Chlorella vulgaris*, and *Raphidocelis subcapitata*. The results showed that the commercial Roundup formula was more toxic than pure glyphosate and its effects on algal cells occurred at concentrations of 0.1–3.7 mg·L^−1^ [68]. Comparing the effects of commercial Roundup^®^ Classic Pro, glyphosate acid, IPA salt of glyphosate, and the surfactant POEA on the two algae *Selenastrum capricornutum* and *Skeletonema costatum*, the median growth inhibition concentration results show that the order of toxicity of the tested chemicals was, from the most toxic to the least, as follows: POEA (IC_50_ = 3.92 mg·L^−1^) > Roundup^®^ Classic Pro (IC_50_ = 5.81 mg·L^−1^) > glyphosate acid (IC_50_ = 24.7 mg·L^−1^) > IPA salt of glyphosate (IC_50_ = 41.0 mg·L^−1^) for freshwater algae *S. capricornutum*, and Roundup^®^ Classic Pro (IC_50_ = 1.85 mg·L^−1^) > glyphosate acid (IC_50_ = 2.27 mg·L^−1^) > POEA (IC_50_ = 3.35 mg·L^−1^) > IPA salt of glyphosate (IC_50_ = 5.89 mg·L^−1^) for marine algae *S. costatum*. These results were adjusted to account for the fact that the toxicity of glyphosate acid was mainly due to its low pH [30]. The different susceptibility of the different species of algae to herbicides is also evidenced by the results of the tests with *Scenedesmus obliquus* and *Chlorella pyrenoidosa*. The EC_50_ for *S. obliquus* was determined to be 55.85 mg·L^−1^ for glyphosate and 26.54 mg·L^−1^ for fluroxypyr, whereas for *Ch. pyrenoidosa*, it was 3.04 mg·L^−1^ for glyphosate and 3.53 mg·L^−1^ for fluroxypyr [69]. Conversely, according to Vendrell et al., *Scenedesmus* species are more sensitive to glyphosate than *Chlorella* species, with EC_50_ values for each species determined as follows: *Scenedesmus acutus* at 24.5 mg·L^−1^, *Scenedesmus subspicatus* at 26.0 mg·L^−1^, *Chlorella vulgaris* at 41.7 mg·L^−1^, and *Chlorella saccharophila* at 40.6 mg·L^−1^ [70]. The literature regarding AMPA indicates the EC_50_ value to be greater than 160 mg·L^−1^ and the toxic ratio of glyphosate to AMPA as TR is 0.27 [71].

When testing the expected environmental concentration (EEC) of triclopyr (2560 mg·L^−1^) on green algae *Scenedesmus quadricauda* and *Selenastrum capricornutum*, 13% growth inhibition and 24% growth stimulation were measured, respectively [72]. For the herbicide fluroxypyr, its toxicity to green algae was assessed using the unicellular alga *Chlamydomonas reinhardtii*, with low concentrations of fluroxypyr (0.05–0.5 mg·L^−1^) found to stimulate its growth and high levels (0.75–1.00 mg·L^−1^) found to inhibit its growth [73]. However, in the case of the herbicide Garlon New, synergistic effects of triclopyr combined with fluroxypyr may bring about different results.

Several studies have confirmed that commercial Roundup^®^ Classic Pro is more toxic to freshwater algae than its active ingredient glyphosate, as the inherent toxicity of the latter is augmented by additives in the former. Furthermore, the toxicity of these herbicidal substances to algae significantly depends on both the concentration of the herbicide and also on the species of algae [67].

### 3.2. Duckweed Acute Toxicity Test

The observed doubling time in the control was 2.4 days on average. The pH of the control samples increased from an initial value of 5.5 to an average value of 5.9. Figure 3 shows a comparison of the inhibitory behavior of the tested herbicides. To make the charts more readable, there is a table in the Appendix A showing the inhibition values corresponding to the concentrations used. Their growth-inhibitory effect was determined to be 100% at the ready-to-use dose for both herbicides. It is most notable that absolute growth arrest was observed for the next five most concentrated solutions of Roundup^®^ Classic Pro, namely: (18,381; 13,875.7; 9190.5; 4595.2; 2297.6; and 1148.8) µg·L^−1^, and for the next seven most concentrated solutions of Garlon New, namely, (15,000; 11,250; 7500; 3750; 1875; 937.5; 468.75; and 234.3) µg·L^−1^. From this point of view, Garlon New is more toxic to *L. minor* than Roundup^®^ Classic Pro, because it shows higher toxicity even at lower concentrations. The highest concentration of Roundup^®^ Classic Pro at which growth inhibition was not total was 574.4 μg·L^−1^. However, the growth inhibition was still at a high level, up to 94%. For Garlon New, the highest concentration at which 100% inhibition was not yet observed was determined to be 117.1 μg·L^−1^, causing 73% inhibition of duckweed growth. Except for the lowest tested concentration of 58.5 μg·L^−1^, Garlon New was more toxic than Roundup^®^ Classic Pro at all concentrations. At the lowest tested concentration, which corresponds to 0.3% of the ready-to-use dose, the inhibition of Roundup hindered duckweed growth by 55.3%, compared to 42.5% for Garlon New. The results of our work show that both tested herbicides, Roundup^®^ Classic Pro and Garlon New, are toxic to the non-target organism *L. minor* at ready-to-use concentrations.

To better evaluate AMPA as the direct metabolite of glyphosate, the concentrations for Roundup^®^ Classic Pro were recalculated to reflect only its glyphosate content. The results show that AMPA has a lower toxic effect on *L. minor* than Roundup^®^ Classic Pro and Garlon New; nonetheless, according to the results obtained from the Kruskal–Wallis test, this difference was not statistically significant. At a concentration of 10.69 mg·L^−1^, duckweed growth inhibition for AMPA was 36.83%. The results of the toxicity tests for AMPA and AMPA with pH adjustment are presented in Figure 4 and Figure 5, respectively. When added to Steinberg’s growth medium, the test substances lowered its pH, with the most significant decrease brought about by AMPA. For this reason, in addition to comparing pure AMPA, pH-adjusted AMPA was prepared as well (Appendix A). After adjusting the pH for the AMPA metabolite, a 10-fold dilution was used. The highest concentration corresponded with molar conversion to the glyphosate content in Roundup^®^ Classic Pro and as such was determined to be 2851 mg·L^−1^. At this concentration, duckweed growth inhibition for AMPA with pH-adjustment was 95.3%. The measured data shows that after a pH adjustment in the medium with the tested AMPA, the inhibitory effect was lower than when using AMPA with no pH adjustment, but the differences were not significant (I_(c=285.1 mg·L^−1^)_ = 85.5%; I_(c=28.5 mg·L^−1^)_ = 80.4%). Figure 5 shows an increase in the total area of the duckweed fronds after four and seven days in the medium with the pH-adjusted AMPA. The herbicides caused chlorosis and necrosis, with full necrosis caused by concentrations ranging from the ready-to-use dose to a dose of 574.4 µg·L^−1^ of Roundup^®^ Classic Pro, and ranging from the ready-to-use dose to a dose of 117.1 µg·L^−1^ of Garlon New. Some fragmentation of the duckweed fronds was also observed. In dilutions of ready-to-use concentrations of Roundup^®^ Classic Pro and Garlon New below 3% and 0.75% respectively, the fronds were not entirely dead. Chlorosis was observed and any new fronds were growing smaller in size and in higher numbers. Furthermore, when the concentration of AMPA was lowered, no necrosis occurred; although partial chlorosis was present in all samples, even at the lowest concentration of 10.7 mg·L^−1^. Root growth, with the exception of controls, was not observed or only very stunted roots grew at the lowest concentrations. Figure 6 shows a comparison of a duckweed colony at the beginning of the test and after seven days, when all *L. minor* fronds turned white.

The values of total chlorophyll in the fronds of the duckweed corresponded to the loss of biomass. None of the measured values exceeded the control samples. In contrast, at the four highest concentrations, no chlorophyll was detected after applying any of the three test substances. When diluted to 12.5%, a small amount of chlorophyll remained in the AMPA samples, and even at the lowest tested concentration, the decrease caused by AMPA was the least severe. The results of the chlorophyll content test are shown in Figure 7.

Our conclusions are in agreement with Sikorski et al., who also determined that glyphosate is toxic to the non-target organism *L. minor*, even at low concentrations. In addition, the authors point to the absorption of glyphosate by duckweed, where after the exposure to 3 μmol·L^−1^ of glyphosate for seven days, glyphosate content exceeded the acceptable maximum residue level ten times [74]. Another work also points to a reduction in biomass and total chlorophyll content in *L. minor* after the application of Roundup^®^ Classic Pro (concentrations of 3.16–31.58 mmol·L^−1^). Here, Roundup^®^ Classic Pro also caused an excessive accumulation of putrescine, spermidine, and total polyamines in the tissues of the duckweed [75].

The harmful effects of glyphosate on *L. minor* are also reflected in photosynthesis, respiration, and pigment concentrations in connection with the oxidative stress caused by the accumulation of hydrogen peroxide by this herbicide [76]. Bioassays revealed that metribuzin was more toxic than glyphosate to *L. minor* [50]. The herbicide flurochloridone causes significant oxidative damage, photosynthetic pigments damage, and bleaching of *L. minor* already at a concentration of 20 μL·L^−1^ of the herbicide [77]. When comparing individual duckweed species, it was found that *Lemna gibba* shows a higher sensitivity to Roundup^®^ Classic Pro and triclopyr than *L. minor* or algae species [78,79].

According to EFSA, glyphosate and its metabolite AMPA are low risk for organisms living in surface water [66]. On the other hand, EFSA assessed that fluroxypyr is harmful to aquatic organisms for both groundwater and surface water, but the risk to organisms is low [80].

### 3.3. The Determination of Dehydrogenase Activity in Soil

A preliminary analysis of our soil sample showed that it was slightly acidic, with pH ~ 6.34. The moisture of the soil was measured to be 72.27%. Ignition losses were determined at 15.84%. The total average sulfur content was set at 0.14% of the soil weight, hydrogen at 1.04%, nitrogen at 0.78%, and the total average carbon content was at 6.45%.

Figure 8 shows a comparison of DHA for Roundup^®^ Classic Pro and Garlon New herbicides. The largest difference between the tested herbicides was observed at a ready to use dose (100%), where the observed activity was 2.3× higher with Garlon New than with Roundup^®^ Classic Pro. At all doses tested, DHA values were higher for Garlon New. In contrast to Roundup^®^ Classic Pro, the highest DHA was observed when the Garlon New herbicide was applied at the lowest dilutions (12.5% and 6.2%). At these low doses, which corresponded to 1.8 mg·L^−1^ and 0.9 mg·L^−1^ of test substance Garlon New, this herbicide stimulated activity to reach levels higher than the control. Figure 9 shows the effect of AMPA on DHA. The highest DHA was observed at the lowest dose tested, which corresponded to a dilution of 6.2%. The DHA value of the control was not exceeded at any tested dose of AMPA.

The results of the ANOVA test indicated a statistical difference between the two compared herbicides and AMPA. Based on the post-hoc Tukey–Kramer multiple comparison test, the difference between Roundup^®^ Classic Pro and Garlon New was significant, which corresponds with the results of the DHA inhibition test. As for the comparison between Roundup^®^ Classic Pro and AMPA, and between AMPA and Garlon New, the difference in DHA inhibition was not statistically significant.

A previous study has reported that glyphosate in doses of 1 µg × g^−1^ of soil reduced enzymatic activity by 16–20%, while a 10 µg × g^−1^ dose reduced enzymatic activity by 24–80% in comparison to the control [56]. At the highest tested AMPA concentration of 2852 mg·L^−1^, which corresponds to 100% of the measured concentration, the determined DHA was 1.6× higher than in the samples treated with Roundup^®^ Classic Pro. This trend was maintained at all concentrations tested (in the range of 1.2–1.6). All values for both substances were lower than that of the control sample. According to another study, DHA was also reduced after the treatment of soil with active substances (glyphosate, metribuzin, diuron, oxyfluorfen) [81].

The effect of glyphosate on DHA was also investigated by Zabaloy et al., who used 10× higher doses of glyphosate than the recommended application rates. However, the results on two different soil samples obtained from different areas did not show consistency. At the beginning of the test, the DHA values were low in both the treated soil samples and the control; after three weeks, the DHA in the glyphosate-treated soil was 63% higher than in the control. On the contrary, in the second soil sample, it was 48% lower compared to the control [82].

Soil pH strongly influences the adsorption of glyphosate; the lower the pH, the higher the adsorption of glyphosate [83]. However, not only pH and soil type, but also the incubation time influence the measured DHA. In our case, DHA was measured after twenty-four hours. According to EN ISO 23753-1 [61], measuring DHA within less than four hours of the application of a herbicide will yield significantly lower results, whereas measuring the same at over six hours of the application will yield slightly increased results. Lupwayi et al. reported that DHA decreased with the increasing frequency of glyphosate-resistant crops in both rhizosphere and bulk soils [84]. Other factors that may affect DHA results include the depth of the soil profile, temperature, organic matter content, soil moisture, soil aeration state and the presence of pollutants [85]. The presence of glyphosate in the soil can cause changes in DHA and in the total content of microorganisms [86]. However, the effects of herbicides on soil microbial biomass and enzymes have been shown to be short-term and, over time, a recovery of biological parameters can be observed [55]. According to EFSA, the persistence of fluroxypyr in soil is on the low to moderate level and its risk for soil organism was assessed as low [80]. When herbicides are used in the environment, their effects can be mitigated. On the other hand, frequent applications of glyphosate-based herbicides can reduce the topsoil layer due to increased soil erosion by reducing vegetation, while deeper soil provides a buffer against the harmful effects of glyphosate on biomass [87].

We plan to focus our further research on testing the effects of herbicides and their combinations on non-targeted algal and plant organisms and we would like to observe these effects in conditions closer to the real-life environment.

## 4. Conclusions

Our work showed that the herbicides Roundup^®^ Classic Pro (active ingredient glyphosate) and Garlon New (active ingredients triclopyr and fluroxypyr), as well as AMPA, being a metabolite of glyphosate, cause morphological changes in non-target organism *L. minor*. Furthermore, these substances exhibited toxicity to both duckweed *L. minor* and freshwater green algae *D. subspicatus* after short-term exposure, not only in application doses but also in much lower concentrations. In both tests with aquatic organisms, Garlon New showed higher toxicity than Roundup^®^ Classic Pro. When evaluating the effect of these substances on the DHA of soil, the level of activity after the application of the test substances was shown as follows: Garlon New > AMPA > Roundup^®^ Classic Pro.

## Figures and Tables

**Figure 1 plants-09-01203-f001:**
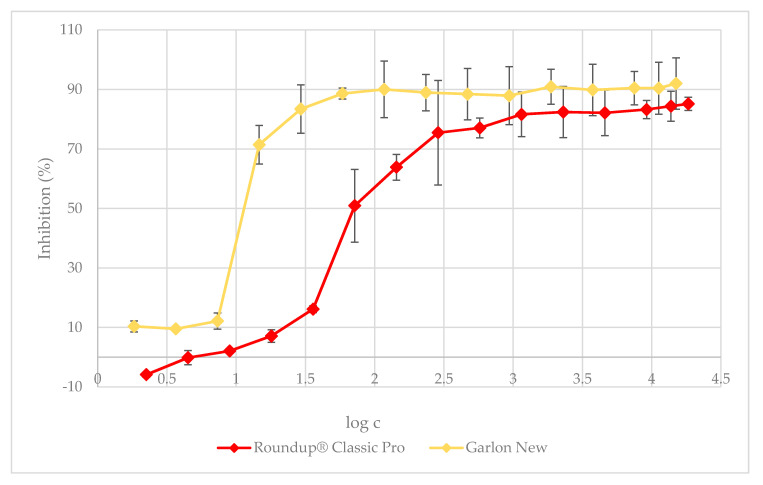
The 72 h algal growth test: the dependence of *D. subspicatus* growth inhibition on Roundup^®^ Classic Pro and Garlon New herbicide concentrations, respectively. The data is expressed as the mean values ± their respective standard deviations.

**Figure 2 plants-09-01203-f002:**
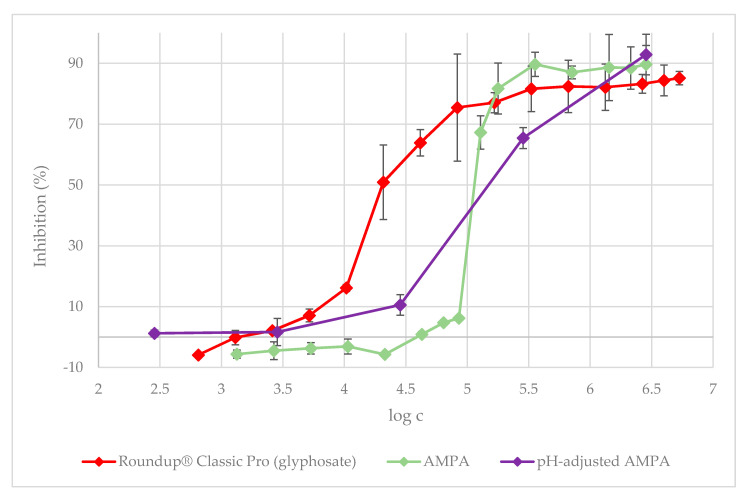
The 72 h algal growth test: the dependence of *D. subspicatus* growth inhibition on Roundup, aminomethylphosphonic acid (AMPA), and pH-adjusted AMPA concentrations, respectively. The concentration of Roundup is represented as the concentration of its glyphosate content. The data is expressed as the mean values ± their respective standard deviations.

**Figure 3 plants-09-01203-f003:**
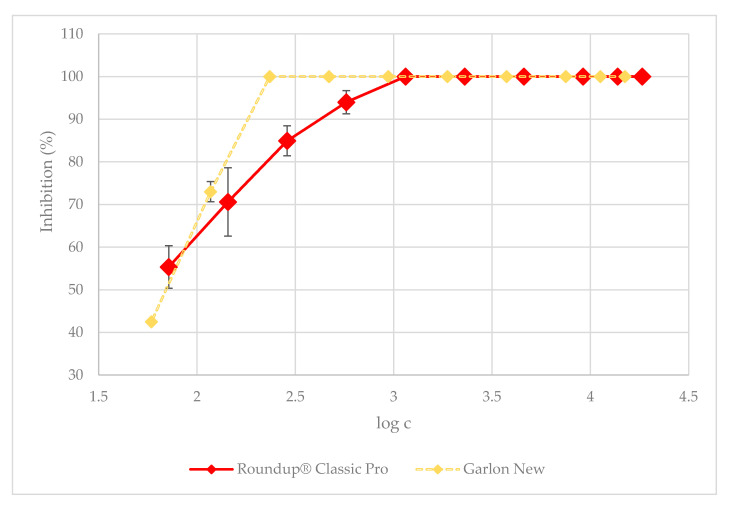
The 7-day duckweed growth test: the dependence of *L. minor* growth inhibition on herbicide-related chemical concentrations. The data is expressed as the mean values ± their respective standard deviations.

**Figure 4 plants-09-01203-f004:**
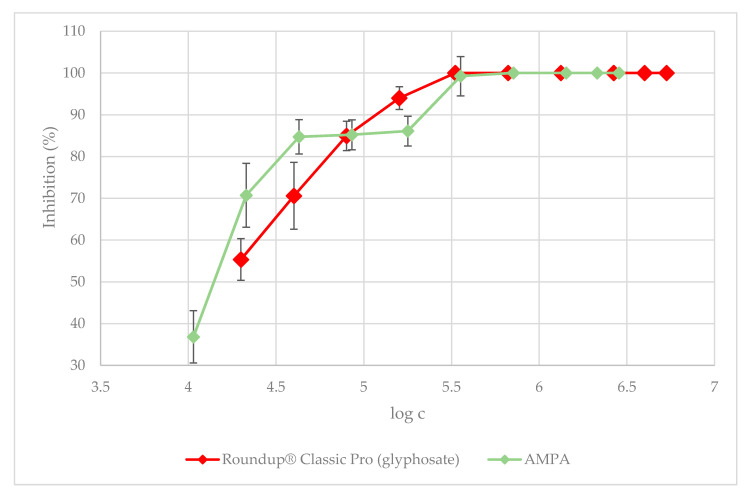
The 7-day duckweed growth test: the dependence of *L. minor* growth inhibition on herbicide-related chemical concentrations. The data is expressed as the mean values ± their respective standard deviations.

**Figure 5 plants-09-01203-f005:**
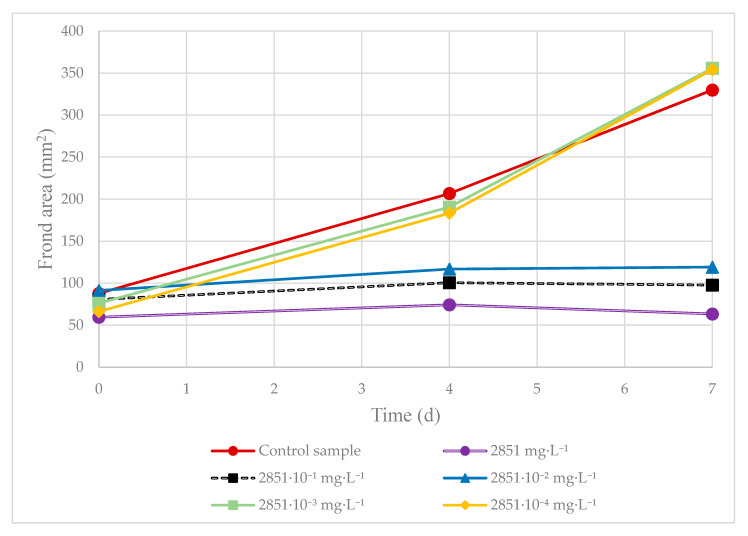
The 7-day duckweed growth test: the dependence of *L. minor* frond area growth in a medium with pH-adjusted AMPA on time.

**Figure 6 plants-09-01203-f006:**
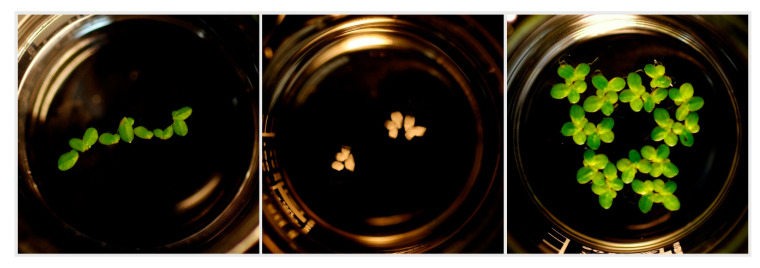
Images of beakers with duckweed *L. minor*. From left to right: the start of the duckweed inhibition test; the end of the test: AMPA (c = 2139 mg·L^−1^); the end of the test: control sample.

**Figure 7 plants-09-01203-f007:**
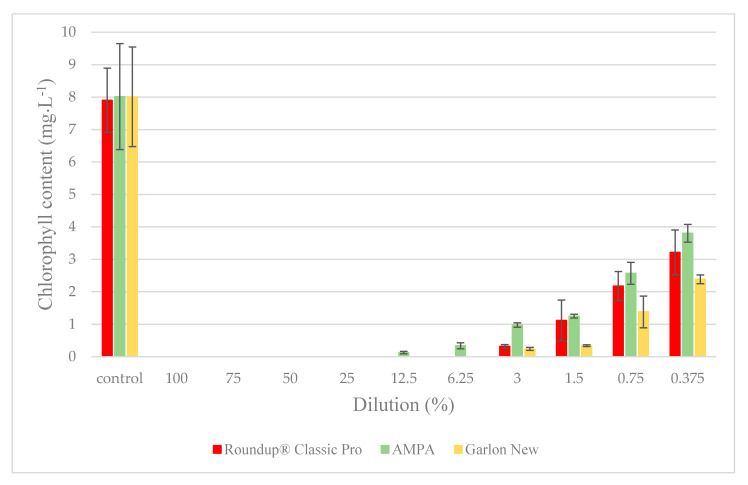
The dependence of the total chlorophyll content in the fronds after 7 days of a duckweed growth test on the dilution percentage of the tested solutions. The data is expressed as the mean values ± their respective standard deviations.

**Figure 8 plants-09-01203-f008:**
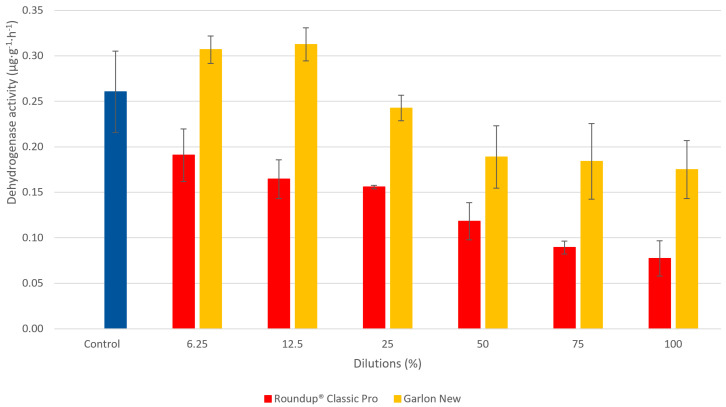
The dependence of dehydrogenase activity (DHA) on the concentrations of the test substances. The ready-to-use dose of herbicides Roundup^®^ Classic Pro and Garlon New was considered 100%. The data is expressed as the mean values ± their respective standard deviations.

**Figure 9 plants-09-01203-f009:**
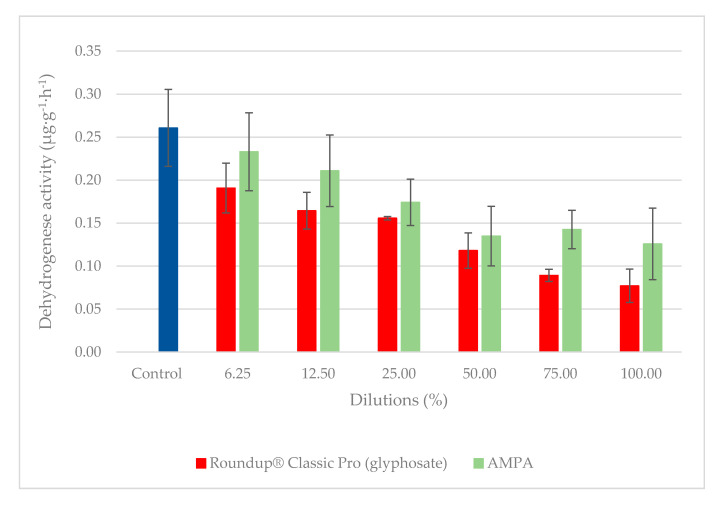
The dependence of DHA on various dilutions of AMPA and Roundup^®^ Classic Pro. The highest AMPA concentration corresponded to the molar equivalent of glyphosate salt present in the Roundup^®^ Classic Pro formula, which is 2852 mg·L^−1^. The data is expressed as the mean values ± their respective standard deviations.

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
