# Peer review of "Determination of the Ecotoxicity of Herbicides Roundup® Classic Pro and Garlon New in Aquatic and Terrestrial Environments"

_plants, 2020, doi:10.3390/plants9091203_

Round 1
Reviewer 1 Report
The submitted report provide aditional data on the toxicological effect of plant protection products (PPP) based on glyphosate and triclopyr-fluroxypyr on algae and aquatic plant.
Authors should take into consideration that the expected concentration in the soil & aquatic environment is not the ready to use dose. There are several factors that have an influence in the estimated concentration of the active substances (AS) in the diferent environmental compartments. The risk assessment of PPP is based on the agronomic use of these type of products and mitigation measures are implemented by the farmers to reduce the exposure. As it is indicated in the abstrac and it is documented the use of PPP may negatively impact the life of non-target organisms, but also this impact is mitigated using different risk mitigation measures.
Authors should justify in the aim of the work the reason to compare the effect of the three active substances (use of the PPP, crops, weeds, market....). It would be preferably to use PPP containing only one active substance for the comparation in order to compare the effect of each AS solely, avoinding any posible sinergistic effect.
Roundup® Classic Pro containing glyphsate contains POEA as a coformulant and the sinergist effect of this coformulant with the active susbtance should have been considered. In the same way the plant protection product Garlon New, that contains a mixture of active substances triclopyr and fluroxypir, the sinergist effect of both active substances should have been taken into consideration and should be discussed.
Authors should take into consideration the EFSA concluision on the peer review of the pesticide risk assessment of the active substance Glyphosate, Triclopyr and Fluroxipyri in which there are data on the degradation in soil and water and toxicity of these active substances. And compare their results with the results used for the european risk assessment. It would be preferably to express the concentration as mg (active ingredient)/L. It is not clear in the text if the concentrations are expresed as active ingredient.
Tests were performed according ISO guidance, that are similar than OECD guidance. OECD criteria for validation of the test should be taken into consideration and the necesary data should be included in the results, as the increased biomass concentration in the control cultures , coefficient of variation daily growth rates in the control cultures during the course of the test that must not exceed 35% in the case of the algae growth inhibition test, etc....
Reviewer 2 Report
Review of the manuscript “Determination of the Ecotoxicity of Herbicides Roundup and Garlon in Aquatic and Terrestrial Environments”
The authors of the manuscript made an attempt to determine the effect of Roundup® Classic Pro, Garlon New and AMPA on aquatic organisms duckweed (Lemna minor) and green algae (Desmodesmus subspicatus), as well as on the enzymatic activity of dehydrogenase in soil.
The paper presents quite interesting results and the research concerns active substances of big economic importance however it’s not ready for publication. The objective of the study was formed clearly. The title is adequate to the topic of research however full names of herbicides should be used. The section “Materials and methods” should be carefully corrected because some information is lacking and/or the information are not precise. Lack of full name of the herbicide Garlon in the title and in some other places in the text can create misunderstanding (Garlon New contains triclopyr and fluroxypyr, Garlon – only triclopyr), active ingredient concentration log transformed below some figures is not clear, it makes difficult to check the results. Figure should be more extensively described in the text. English should be revised. Therefore I recommend major revision of the manuscript.
Title
Full names of herbicides should be used. Garlon has one active ingredient and Garlon New – two a.i.
Introduction
Page 1 line 37-38 - the authors should write: The most widely used herbicides instead of pesticides. Fungicides and insecticides are also frequently used.
Line 76 – systemic not systematic should be used
Line 109 – Atrazine, paraquat and glyphosate are active substances of herbicides, not herbicides. What is primeextra?
Results and Discussion
Lines 121, 124-125 - it is unnecessary to give once again the composition of Roundup Classic Pro and Garlon New. It should be written in Material and methods section, which should be placed before the Results and Discussion section.
Lines 182-184 comparison of the toxicity of Roundup versus glyphosate wasn’t investigated by the authors therefore this summary is not necessary.
Lines 188-189 – the section 2.2 shouldn’t start from the conclusion. The authors should first comment Fig. 3.
Figures 1-3 The values of herbicides concentrations are log transformed – it is a very uncomfortable way to read and interpret such figure. It should be clearly explain what log c value correspond to which herbicide or AMPA concentration. Why the unit in Fig. 1 is μg·L-1 and in the corresponding part of Fig. 3 we can see μl·L-1 . The results presented on figures should me more extensively described and discussed in more detail in the text.
Similar results (inhibition of growth) for algae are divided into two figures (Fig 1 and 2) and results of growth inhibition of duckweed are presented on one figure. The authors should decide to one way of results presentation.
Lines 265-266 The phrase “a comparison of the influence of 265 Roundup and Garlon with a different mode of action and Roundup with AMPA (a direct metabolite 266 of glyphosate) on DHA in soil” should be rewritten.
Lines 287-288 these are active substances, not pesticides. Metam Na is a mistake.
Materials and methods
Materials and methods section should be more precise.
“The highest concentration of the herbicides tested was their ready-to-use concentration of 15 mL·L-1. The next highest tested concentrations were 11.25 360 mL·L-1 and 7.5 mL·L-1. The subsequent concentrations were prepared by repeated two-fold dilution”(…) “In the pH-adjusted assay, decimal dilution was used and five concentrations plus a control were tested”
Please, provide the numbers.
Section 3.3.3 line 401 – what is “real soil”? The soil type should be described. Also the organic matter content should be given because it is a very important factor influencing adsorption of herbicides in soil.

Round 2
Reviewer 2 Report
I would like to thank the authors for their effort to improve the manuscript. I accept the corrections and the manuscript in the present form.